# Evolutionary Method of Heterogeneous Combat Network Based on Link Prediction

**DOI:** 10.3390/e25050812

**Published:** 2023-05-17

**Authors:** Shaoming Qiu, Fen Chen, Yahui Wang, Jiancheng Zhao

**Affiliations:** Communication and Network Laboratory, Dalian University, Dalian 116622, China; qiushaoming@dlu.edu.cn (S.Q.); wangyahui@s.dlu.edu.cn (Y.W.); zhaojiancheng@s.dlu.edu.cn (J.Z.)

**Keywords:** heterogeneous combat networks (HCNs), link prediction, operational capability

## Abstract

Currently, research on the evolution of heterogeneous combat networks (HCNs) mainly focuses on the modeling process, with little attention paid to the impact of changes in network topology on operational capabilities. Link prediction can provide a fair and unified comparison standard for network evolution mechanisms. This paper uses link prediction methods to study the evolution of HCNs. Firstly, according to the characteristics of HCNs, a link prediction index based on frequent subgraphs (LPFS) is proposed. LPFS have been demonstrated on a real combat network to be superior to 26 baseline methods. The main driving force of research on evolution is to improve the operational capabilities of combat networks. Adding the same number of nodes and edges, 100 iterative experiments demonstrate that the evolutionary method (HCNE) proposed in this paper outperforms random evolution and preferential evolution in improving the operational capabilities of combat networks. Furthermore, the new network generated after evolution is more consistent with the characteristics of a real network.

## 1. Introduction

The research on network modeling and dynamic evolution characteristics of weapons and equipment is a very important practical issue in information age warfare. Heterogeneous networks provide a suitable model for describing real combat networks. Previous research on combat networks has mainly focused on homogeneous networks, but, in reality, combat networks are composed of a large number of heterogeneous nodes and edges with different functions. So far, there has been little research on HCNs. The network motif is a subgraph pattern that appears more frequently in the network than other structures, which can help understanding of the formation principles or evolving mechanics of complex network [1]. Because of the complex nonlinear relationships and emergence among combat units in a combat network, the implicit pattern information in the combat network can be used to build combat network models and analyze these complex collaborative relationships. In combat networks, collaborative interactions abound. From a combat network perspective, only when certain specific combat units are combined to form a structure can it have higher operational capabilities. Arbitrarily removing nodes or edges from them will reduce operational performance. Multiple combat units collaborate to form an operation chain, which constitutes the combat capability of the entire combat network. Mining subgraph patterns of combat networks can help understand and explain the potential properties of combat networks, providing a new research perspective for combat network modeling, evolution, and recovery. In this paper, we first consider frequent subgraph structures in combat networks. Then, by comparing the differences between subgraphs, we predict the location of new collaborative relationships and combat units that will appear. After that the combat network evolution simulation is conducted. A standardized operational capability metric is applied to evaluate the performance changes of an HCN. The results show that, compared with traditional methods, the evolutionary method proposed in this paper can enable the combat network to evolve towards the direction of increasing operational capabilities, providing a new method for optimizing an HCN, which warrants in-depth research.

The main advantages and contributions of the method proposed in this article are: (1) A new link prediction index is designed for HCNs. The link prediction method proposed in this paper (LPFS) has higher prediction accuracy and performs better than other baseline methods on HCNs. (2) LPFS can not only predict the emergence of new collaborative relationships, but can also predict the emergence of new nodes. This is a problem that traditional link prediction has difficulty solving. (3) A new evolutionary method for combat networks is proposed, which performs better in improving operational capabilities than preferential evolution and random evolution.

The main structure of the article is as follows: Section 2 mainly introduces relevant work; Section 3 presents an HCN model and describes the link prediction algorithm in detail; Section 4 introduces the definition and calculation of combat capabilities, as well as the evolution steps of HCNs; Section 5 analyzes the experimental results and Section 6 is the conclusion and summary.

## 2. Related Work

Complex network theory provides a new perspective and method for the study of war systems and complex systems. Traditional complex networks mainly focus on isomorphic networks (all nodes are of the same type or have the same functionality in the network), while real-world networks, such as author citation network [2], which include papers and authors, and social network [3], which include users and items, are heterogeneous networks composed of different types of entities and relationships. Compared to homogeneous networks, heterogeneous networks contain rich structural and semantic information, which can more accurately describe the complex relationships between entities. The research approach to network evolution is to model and abstract the topological structure of a network based on its key characteristics and real data [4]. A dynamic evolution model is used to replace the real network for research and analysis, focusing on exploring various indicators of the network after evolution under different conditions and times. Finally, the characteristics of the network are recognized and applied to practical construction. This requires a full understanding of the dynamic characteristics of networks. Researchers hope to explore the rules of network evolution by analyzing the evolution and dynamic processes of networks through data analysis and other means. The rationality of the evolution rules needs to be judged by testing whether they can provide more accurate link prediction. The research on combat network evolution aims to reveal the network dynamics mechanism during the evolution process. It uses simulation networks generated by network modeling to study real combat systems, and to explore and analyze their generation and evolution mechanisms. The key lies in the fact that the constructed combat system network model must truly reflect the realistic characteristics of the combat system, such as its composition and connection rules. Care [5] first used complex network methods to establish combat models in the information age. Dekker [6] simulated and analyzed the impact of different network types, such as regular networks, small-world networks, and scale-free networks on operational capability. Then there was further study on combat networks, including the robustness of combat network [7], and operational capability analysi [8,9].

Link prediction predicts future links or missing links based on observed information [10,11]. Most traditional link prediction algorithms are based on homogeneous networks, where the nodes and links are of the same type. Directly applying these methods to complex heterogeneous networks will lose a lot of information [12]. Currently, the application of link prediction in military applications is mainly based on the obtained enemy network topology, predicting or identifying key nodes in advance. If the prediction results are accurate, it will be conducive to optimizing the organizational structure of our military and attacking the key operational forces of the other party [13]. There is the difficulty, or even the impossibility, of obtaining a completely accurate military network. This means that enemy combat networks are usually incomplete and sparse. So, using traditional methods to predict combat network topology will fail. Therefore, some scholars have proposed a meta-path-based link prediction method that can simultaneously predict multiple types of connection [14,15]. Meta-path-based methods can utilize not only the observed structural link information but also the semantic information of meta-path features. Meta-path features reflect the basic connectivity of object pairs and are key to the mining and learning process of general heterogeneous networks. The selection of path characteristics determines the effectiveness of link prediction algorithms, which requires a sufficient understanding of network connectivity characteristics. Due to the high computational complexity, it is difficult to enumerate all meta paths for a given HCN. Existing research only considers meta paths with lengths of 2 and 3, ignoring the higher-order structures in the network. Although the meta-path-based approach achieves simultaneous prediction of different types of links, it is achieved by predicting a certain type of link separately.

The main method for studying the evolution of combat networks is to directly establish evolutionary models to speculate on the factors that affect combat networks. Due to the power law distribution of node degrees in combat networks, and the priority connection based on node degrees which can generate scale-free network [16], the degree priority evolutionary model is currently the main method for generating combat networks. However, the explanation of connection mechanisms in these models is relatively simple. Real complex networks exhibit many different attribute characteristics, which indicates that the connection mechanism of a network is likely to be determined by multiple mechanisms rather than a single one. Currently, there is also a lack of unified evaluation criteria. It is also unclear how these mechanisms collectively affect the evolution and capability of combat networks. This article uses the link prediction method to study the connection mechanism of combat networks. If the principle of the link prediction algorithm is consistent with the connection mechanism of the formation and evolution of combat networks, it can provide accurate predictions and can evaluate the connection mechanism based on the prediction results. Using link prediction to reveal network evolution mechanisms has been applied in trading network [17] to study the impact of different factors. However, as far as we know, there are few related applications in heterogeneous networks.

In this paper, the method of mining frequent subgraphs is used to discover features in combat networks. In a combat network, the capabilities of the combat system are composed of multiple operational chains. The capability of a single combat unit in a combat network is difficult to measure, and multiple combat units collaborate to form an operational chain, which constitutes the capability of the entire combat network. Studying the frequent subgraphs that appear in the combat network can indicate the evolution rules of the macrostructure from the microstructure, providing a new reference for the optimization of the combat network structure. Firstly, this utilizes the known operational network structure and complete data. Secondly, subgraphs can consider more hierarchical structures. There are many classical algorithms for mining frequent subgraphs, such as Grami [18], gSpan [19], etc., which also have good operational efficiency on large networks. Another challenge for network evolution and connectivity mechanisms is how to add constraints or control factors to the acquired evolution mechanisms to achieve the desired functional performance. This paper involves the design of different experiments to add edges and nodes to the network and calculates the changes in operational capabilities of the combat network during this evolution process.

## 3. Link Prediction Based on Frequent Subgraphs

Modern combat cycle theory suggests that the combat process is a cyclic process composed of observation, orientation, decision, and action (OODA) [20,21]. That is, the search node discovers the enemy target and transmits information to the decision node. After detailed analysis, the decision node transmits commands to the influence node. After receiving the commands, the influence node attacks the target. Therefore, according to the functions of combat units, they are divided into the following categories: (1) Search Node(S): Weapon equipment that uses sensors to collect target and battlefield information. Its main functions include target reconnaissance, intelligence acquisition, and battlefield surveillance. (2) Decision Node(D): Refers to weapons and equipment that have the functions of information processing and analysis, assisting decision-making, and implementing command and control over interfering entities. (3) Influence Node(I): Refers to the equipment entity that mainly performs operational damage operations, with specific functions, such as precision strike, fire damage, and electronic jamming.

Define an HCN as G={V,E,L,M}, *V* is a set of nodes and *E* is a set of edges, M={D,I,S} is a set of node types, and L={S→D,S→I,D→I,I→I,D→D} is a set of edge types. Figure 1 is a heterogeneous combat network, with red representing search nodes, green representing decision nodes, and blue representing influence nodes.

### 3.1. Link Prediction Index Design

This paper proposes a link prediction method based on frequent subgraphs. Firstly, frequent subgraphs must be mined on HCN. The frequent subgraph mining algorithm can obtain all frequent subgraph patterns that meet the support threshold from the graph. gSpan [19] is currently one of the most effective frequent subgraph mining algorithms in terms of mining efficiency and scalability, so it is used in experiments. Then compare the differences between different subgraphs. If only one edge (such as g1 and g2 in Figure 2) differs between two subgraphs, record the nodes and edge type. If two subgraphs differ by one node and one edge (such as g3 and g4 in Figure 2), record the missing node type and edge type. Algorithm 1 shows the specific steps. Firstly, the gSpan algorithm is used to mine frequent subgraphs that meet the threshold. As shown in Figure 2, g1, g2, g3, and g4 are frequent subgraphs in G, and g1 and g2 are isomorphic subgraphs that differ by one link, and g3 and g4 are isomorphic subgraphs that differ by one edge and one node. If their occurrence times are 8, 6, 10, and 6, respectively, then score (edge(3, 4) = 0.75, score(edge (nodeType (3), 2)) = 0.6.
**Algorithm 1** Link prediction based on frequent subgraphs (LPFS)**Input:** *G*, min_sup, max_size**Output:** scores1: frequent_sub=gSpan(G,min_sup,max_size)2: **for** sub1,sub2 in frequent_sub
**do**3:    **if** isomorphism(sub1,sub2,max_n=1)
**then**4:      node1,node2,edge_type=find_edge(sub1,sub2,G)5:      scores[node1,node2,edge_type]+=count(sub2)/count(sub1)6:    **end if**7: **end for**8: **return**  scores

### 3.2. Link Prediction Evaluation Indicator

AUC (area under the receiver operation characteristic curve) is an indicator that measures the accuracy of link prediction algorithms in predicting the overall network, and is also the most widely used one. For example, its main content is to divide the set *E* into a training set $E^{T}$ and test set $E^{P}$ according to a certain proportion. $\bar{E}$ represents a nonexistent set of edges, $U=E^{P}+E^{T}+\bar{E}$. The algorithm is executed in $E^{T}$ to obtain a predictive value matrix, with each edge $e_{ij}\in \bar{E}\cup E^{P}$ obtaining a predictive value score(i,j). Randomly select an edge from $E^{P}$ and $E^{T}$, compare their scores, and repeat for *n* times, $AUC=(n_{1}+0.5n_{2})/n$. ( $n_{1}$ is the number of times that the score of $E^{P}$ is higher than the score of $\bar{E}$, and  $n_{2}$ is the number of times when the scores are equal). The larger the AUC value, the higher the accuracy of the algorithm’s prediction.

## 4. Evolution of Heterogeneous Combat Networks

The purpose of the design of link prediction indicators is to reveal the mechanism of network evolution, find the evolution method that best matches the characteristics of heterogeneous combat networks, and to establish a combat network evolution model. A new combat network is created that conforms to the structural characteristics of a real combat network through the establishment of collaborative relationships between combat unit entities in the network and the addition of combat unit entities. The network can then exhibit higher operational capabilities.

### 4.1. Heterogeneous Combat Network Operational Capabilities

In order to accomplish a specific combat mission, combat units with different functions collaborate to form an operational chain. The basic operational process of the operational chain is described as follows: The search node transmits the collected data and information to the decision node, which processes and analyzes it, and then issues a command to the influence node. The greater the number of operational chains, the stronger the operational capabilities of the operational network. At the same time, the length of the chain affects the efficiency of information transmission, and the shorter the battle chain, the higher the efficiency of information transmission. In this paper, the operational chain is defined as a path from the search node to the influence node, and this path includes at least one decision node. Therefore, how to calculate the number and length of operational chains is a key issue for operational capabilities calculation. The detailed steps are shown in Algorithm 2. max_len represents the maximum length of the path. Because the information transmission efficiency of an overly long chain is not high, the length of the chain is limited to 5.
**Algorithm 2** Calculate operational networks ability (Cal_ability)**Input:** *G*, max_len**Output:** ability  1: I=G.node(nodetype==1)  2: S=G.node(nodetype==2)  3: D=G.node(nodetype==3)  4: **for** *s* in *S*
**do**  5:    **for**
*i* in *I*
**do**  6:      **for** pathinall_simple_path(G,s,i,cutoff=max_len)
**do**  7:         **if** size(D.intersection(path))>=1andsize(I.intersection(path))==1
**then**  8:           ability+=1/len(path)  9:         **end if**10:      **end for**11:    **end for**12: **end for**13: **return**  ability

### 4.2. Evolution of Heterogeneous Combat Networks

In order to study which evolutionary method can improve the operational capability of the combat network most rapidly, we designed two experiments, one involving only the establishment of collaborative relationships between combat unit entities, and the other involving the addition of new combat units. The specific experimental steps are shown in Algorithms 3 and 4. The establishment of cooperative relationships in combat networks is manifested as the increase in edges. The edge_num is the number of edges to be added. First, run LPFS on a combat network, and some unobserved connections in the network will receive a score. The larger the score, the more likely or necessary the connection is to be generated. Each time the LPFS is run, an edge with the largest score is added to the network. Finally, we obtain a new network and its operational capability after adding each new edge.
**Algorithm 3** Heterogeneous Combat Networks Evolution (HCNE) add_edge**Input:** G,edge_num,max_len**Output:** new_G,Ability  1: scores=LPFS(G,min_sup,max_size)  2: sort(scores,ascending=False)  3: **for** *i* in edge_num
**do**  4:    **for** score in scores
**do**  5:      node1,node2,edge_type= scores.key()  6:      **if** !G.has_edge(edge(node1,node2,edge_type))
**then**  7:          G.add_edge(edge)  8:          Cal_ability(G,max_len)  9:          scores=LPFS(G,min_sup,max_size)10:         sort(scores,ascending=False)11:         break12:      **end if**13:    **end for**14: **end for**15: **return**  new_G,Ability

**Algorithm 4** Heterogeneous Combat Networks Evolution (HCNE) add_node
**Input:** G,nodes,max_len
**Output:** new_G,Ability
  1: scores=LPFS(G,min_sup,max_size)
  2: sort(scores,ascending=False)
  3: **for** node in nodes
**do**
  4:    degree(node)=Degree(G,nodetype(node))
  5:    **for**
*i* in degree(node)
**do**
  6:      **for** score in scores
**do**
  7:         node1,node2,edge_type=scores.key()
  8:         **if** node1==nodetype(node)
**then**
  9:            G.add_edge(node,node2,edge_type)
10:           Ability.append(Cal_ability(G,max_len))
11:           break
12:         **end if**
13:         **if** node2==nodetype(node)
**then**
14:           G.add_edge(node1,node,edge_type)
15:           Ability.append(Cal_ability(G,max_len))
16:           break
17:         **end if**
18:         scores=LPFS(G,min_sup,max_size)
19:         sort(scores,ascending=False)
20:      **end for**
21:    **end for**
22:  **end for**
23:  **return**  new_G,Ability


If a new combat unit needs to be added to the combat network, it is first necessary to calculate the average degree of this type of node and determine how many nodes in the network should be connected to this node. Then run LPFS and find the most likely location in the network where this type of node is missing. The nodes is the set of nodes to be added. We then obtain a new network and its operational capability at each step of the evolution process. Finally, we obtain a new network and the operational capability value after adding each new node.

## 5. Experiments

The data set selected in this article is from [13,22]. The main topological characteristics of the data are shown in Table 1. |V| and |E| represent the total number of nodes and edges in the network, respectively. |S|, |D| and |I| represent the number of search nodes, decision nodes, and influence nodes, respectively. <k> is the average degree of the network, and <d> is the average shortest path of the network.

### 5.1. Link Prediction

Table 2 lists the AUC average and variance for 100 random partitions of different link prediction algorithms [10,13] when the test set ratio is 10%. These algorithms include similarity based on local information considering common neighbors and node degrees, such as CN, PA; path-based similarity indicators, such as LocalP and Katz; and similarity index based on random walk, such as RWR, SRW. The largest AUC value and the smallest variance value are in bold. It can be seen that the AUC value of LPFS has reached 0.86, which is higher than other methods, and is about 0.09 higher than the optimal LR [13], with a smaller variance. This indicates that the prediction accuracy of LPFS is higher and more stable.

As shown in Figure 3, LPFS outperforms almost all other link prediction algorithms when the proportion of test sets varies. The parameter set in the experiment is max_sup = 6, max_size = 4. When the test set proportion gradually increases and is greater than 30%, the AUC value of LPFS gradually decreases but remains high. When the test set proportion is greater than 30%, the AUC value of LPFS is slightly lower than LR. This is because, as there are more missing edges in the network, frequent subgraphs gradually decrease, and the information utilized by LPFS gradually decreases.

#### Parameter Discussion

Figure 4 shows the impact of min_sup and max_size changes on AUC (the training set ratio is 10%). Each polyline displays the effect of max_size on AUC when min_sup is the same. As can be seen from the figure, when max_ size = 3 or 4, the AUC value is the highest, with max_size gradually increasing and the AUC value gradually decreasing. However, when min_sup = 14, max_size ≤ 7, the AUC value is not high and remains around 0.65–0.7. However, when max_size > 7, AUC has increased and reached around 0.8. It can be concluded that, in a combat network, the pattern formed by 3–4 combat units occupies the dominant position, including most of the information in the combat network. At the same time, higher-order structures in operational networks can also express certain characteristics of operational networks. Therefore, when the LPFS utilizes subgraphs with fewer nodes, max_sup should be set to be smaller to avoid missing information and causing inaccurate prediction results. When using subgraphs with fewer nodes, the max_sup should be set higher, which can filter out much interference information.

### 5.2. Evolution of Heterogeneous Combat Networks

For comparison, random evolution and preferential evolution experiments were designed. When only adding edges, random evolution randomly selects two nodes to connect each time. Preferred evolution considers the node degree when selecting nodes. The greater the degree, the greater the probability of being selected. Figure 5 shows the changes in operational capabilities during the evolution process. In two experiments, we added 100 edges and 100 nodes to the network, respectively. The nodes are selected based on the proportion of different types of nodes in the original combat network. The order in which nodes are added is random. In the case of only adding edges without adding nodes, each additional edge improves the operational capability of the combat network to a certain extent. HCNE has the fastest and largest increase in operational capability, which is superior to preferential evolution and random evolution. When adding a new node to the network, how many old nodes should be connected depends on the average degree *d* of the type of node. Therefore, when randomly evolving to add nodes, the new node is randomly connected to *d* nodes in the network. During the preferential evolution, the probability of a new node selecting an old node is positively correlated with the degree of the old node. As shown in Figure 5, when the number of new nodes is less than 20, the impact of the three evolution methods on operational capabilities is not significantly different. However, when the number of nodes is greater than 20, the advantages of preferential evolution and HCNE gradually emerge, and HCNE is superior to preferential evolution. This advantage is even more evident when the number of new nodes is greater than 50.

Figure 6 shows the node degree distribution of the evolved new combat network. As can be seen, the node degree of the original combat network follows the power law distribution. The new network degree distribution generated by the HCNE is more consistent with the original network degree distribution, with the largest number of nodes having a degree of 4. In addition, the number of nodes with a degree greater than 13 in the network generated by HCNE is more than in other networks, which is also in line with the characteristics of combat networks. After the number of nodes and edges in the network increases, some decision nodes need to command and communicate with more nodes, becoming large-degree nodes in the network.

From the results of link prediction, the proposed method has the highest prediction accuracy. This suggests that our evolutionary model is plausible. In the process of evolution, we have selected the edge with the highest prediction score for connection every time, so each step is in line with the evolution of the combat network. It can also be seen from the changes in combat network combat capability that the evolution method proposed by us shows the greatest improvement in combat capability.

## 6. Summary and Analysis

Studying the evolution of HCNs has important practical significance. Currently, the research on the evolution of HCNs mainly focuses on directly establishing evolutionary models to speculate on the factors affecting network evolution, and it is difficult to quantitatively compare different models. This paper uses link prediction methods to study the evolution of combat networks and proposes a link prediction index based on frequent subgraphs. Considering the high-level interactions and rich node attribute information in combat networks, it provides a new method and perspective for revealing the evolution mechanism of combat networks. Firstly, the prediction accuracy of LPFS is better than that of other baseline methods. Secondly, applying this method to the evolution of combat networks, experiments have shown that adding the same number of nodes and edges can improve combat capabilities. Moreover, the final evolved combat network has the same degree distribution as the original network, which conforms to the characteristics of scale-free networks. This is of great significance for the simulation of real-world networks. Since combat networks are heterogeneous, the method is also applicable to citation networks, social networks and so on. It can also be used to solve cold start problems in recommended systems. In the future, the method should be validated in more datasets.

In addition, we can study more capabilities in the same combat network, such as overall capabilities, command capabilities, and logistics capabilities. However, how to define these capabilities is an important issue that needs to be solved urgently. We also need more real data. The key idea of this article is applicable to the study of network evolution. For example, we can also select a link prediction algorithm suitable for co-authored networks, and study the innovative changes of co-authored networks during evolution.

## Figures and Tables

**Figure 1 entropy-25-00812-f001:**
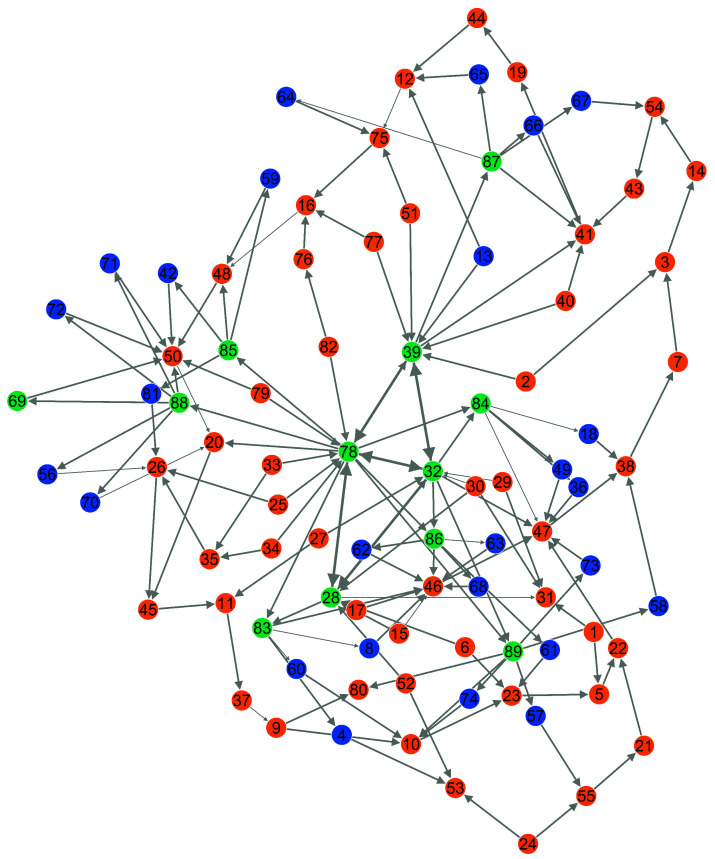
Heterogeneous warfare network (HCN). Red representing search nodes, green representing decision nodes, and blue representing influence nodes.

**Figure 2 entropy-25-00812-f002:**
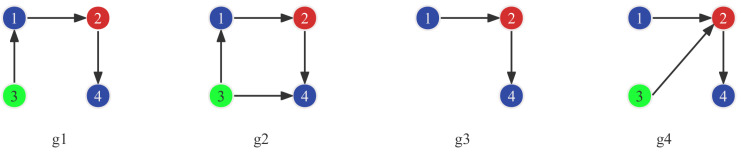
Example of frequent subgraphs with one edge apart.

**Figure 3 entropy-25-00812-f003:**
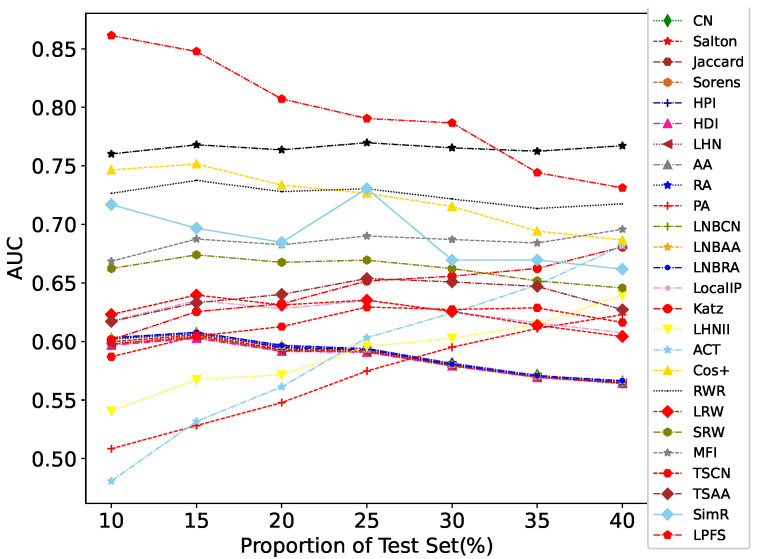
AUC value changes of each link prediction algorithm with different test set proportions. The x-axis represents the proportion of the test set, the y-axis represents the AUC, and each line represents the AUC change of an index at different proportions of the test set.

**Figure 4 entropy-25-00812-f004:**
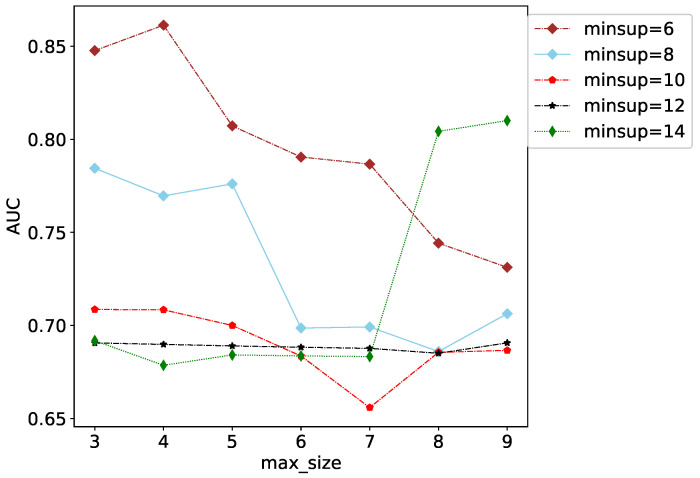
The effect of min_sup and max_size on AUC (when the training set ratio is 10%).

**Figure 5 entropy-25-00812-f005:**
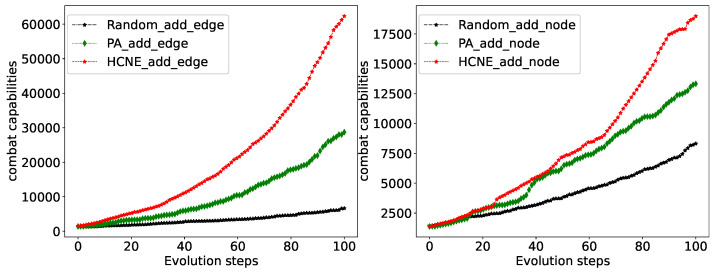
Combat network capability changes using different evolutionary methods. The x-axis represents the evolution steps (adding a new connection or a new node at each step), and the y-axis represents operational capabilities.

**Figure 6 entropy-25-00812-f006:**
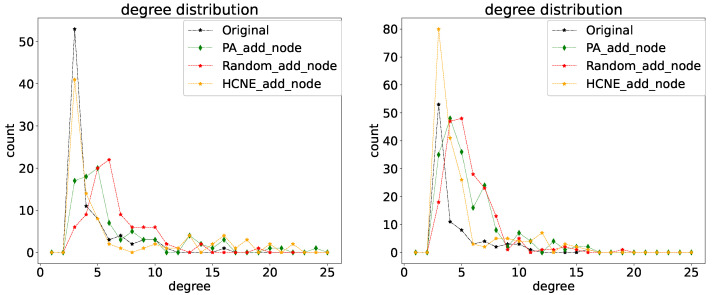
Combat network degree distribution. The x-axis represents the node degree, and the y-axis represents the number of nodes.

**Table 1 entropy-25-00812-t001:** Topological Characteristics of Combat Networks.

|V|	|E|	|S|	|D|	|I|	<k>	<d>
89	150	26	51	12	3.483	8.113

**Table 2 entropy-25-00812-t002:** AUC of link prediction.

**Index**	**LR**	**CN**	**Salton**	**Jaccard**	**Sorens**	**HPI**	**HDI**	**LHN**	**AA**
AUC	0.7771	0.6129	0.6118	0.6093	0.6093	0.615	0.6079	0.6105	0.613
Var	0.0025	0.0038	0.0038	0.0036	0.0036	0.0039	0.0036	0.0037	0.0038
**Index **	**RA**	**PA**	**LNBCN**	**LNBAA**	**LNBRA**	**LocalP**	**Katz**	**LHNII**	**ACT**
AUC	0.613	0.5144	0.6091	0.6091	0.6091	0.6324	0.6235	0.5632	0.4942
Var	0.0038	0.0037	0.0039	0.0039	0.004	0.0046	0.0054	0.0049	0.0053
**Index**	**CosPlus**	**RWR**	**SimR**	**LRW5**	**SRW**	**MFI**	**TSCN**	**TSAA**	**LPFS**
AUC	0.7686	0.7549	0.7355	0.7027	0.6995	0.6944	0.606	0.6375	**0.86135**
Var	0.0020	0.0030	0.0028	0.0036	0.0040	0.0033	0.0057	0.0054	**0.00037**

The best parameters for some indexes are: Katz(0.001), LHNII(0.9), LRW(5), SRW(5), SimR(0.8)

## Data Availability

The data presented in this study are available in [13].

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
