# Peer review of "Evolutionary Method of Heterogeneous Combat Network Based on Link Prediction"

_entropy, 2023, doi:10.3390/e25050812_

Round 1
Reviewer 1 Report
The article touches on an interesting issue. The study of process dynamics in complex networks especially in terms of their heterogeneity can have many applications also in the field of HCNs.
However, the authors should refer more extensively to the issue of complex systems theory for ICT, sensor networks. It may also be interesting to address issues of phase transition in network infrastructure. Phase transition is a natural part of evolution.
The authors refer to the assumptions of complex systems. However, there is no lack of clear indication in the proposed methods/algorithms of the mechanisms characteristic of these complex systems, e.g. emergence, non-extensionality, power law, etc. Have the authors considered including the properties of complex systems more generally?
In addition, authors should consider whether section 3.3 should be made separate?
Is the HCN presented in Fig. 1 a representation of a real system or some random graph?
The authors compare LPFS with other methods without even briefly characterising them. In such an approach, the reader has a difficult time verifying whether the comparison makes sense or whether the assumptions of the individual methods allow for such an operation.
When analysing the results obtained, it should be made clear how the proposed solution improves combat capability.
The authors must indicate their contribution and the proposed approach’s novelty. Has LPFS not been used before in the approach in question?
- No detailed comments.
Reviewer 2 Report
The article titled "Evolutionary Method of Heterogeneous Combat Network Based on Link Prediction" has been submitted to the MDPI Entropy journal for review and possibly to be published. The proposed method of the article uses link prediction methods to study the evolution of heterogeneous combat networks. A link prediction index based on frequent subgraphs (LPFS) is proposed. LPFS has been tested on a real combat network and compared with a number of baseline methods.
I have listed my comments below. I recommend that you answer my questions briefly or provide references. In my opinion, the quality of the manuscript is good already in its current form.
I have the following comments:
1) (Introduction). I suggest that you extend the discussion about operational capabilities. How do you define the concept of operational capability?
2) (Discussion or Experiments). You should have more discussion and explain how the method is related to a wider perspective. Maybe you should add a new section "Discussion". How does your suggested method improve operational capabilities? There are other operational capabilities, for example, logistics. Is it possible to extend the method with other capabilities? Military capabilities are often analysed as hierarchic structures (total capability, capability areas, sub-capabilities etc.). Is the method best suited for investigating a limited set of functionality (for example, "OODA-loop") or is it possible to build a methodology where more functionalities are investigated in the same model? I guess that the extended model would be even more complex and it would be difficult to handle and understand the interdependencies of the model ...
In the section "Related work" on lines 119-123: "In this paper, the method of mining frequent subgraphs is used to discover features in combat networks. In a combat network, the capabilities of the combat system are composed of multiple operational chains. The capability of a single combat unit in a combat network is difficult to measure, and multiple combat units collaborate to form an operational chain, which constitutes the capability of the entire combat network."
In the section "Experiments" on lines 247-248: "At the same time, higher-order structures in operational networks can also express certain characteristics of operational networks."
Can you say something about how your work is related to these aspects?
3) (Introduction or Discussion). Is your method related to other methods used in the field of "Social Network Analysis"? Link prediction and network evolution models are used in analysing social networks.
4) (Related work). I suggest that you include the definition of "isomorphic" because it is not obvious that everybody knows it.
5) (Introduction, Experiments or Discussion). You mention "real combat systems". Can you explain what kind of real combats your model is describing and why the network structures you are using are realistic?
6) Line 76: Decker[? ]
7) Line 79: networks[7], Operational
8) LIne 149: g.is
9) You should provide references for"gSpan" line 156 and line 226 "different link prediction algorithms". Please check if there are other similar parts in the text. Maybe you should say something (briefly) about these link prediction algorithms.
10) Line 165: respectively. Then: score
11) Line 175: edge score of EP larger than
12) Line 188: entities. And
13) Line 197: Algorithm ??
14) Line 215: missing. nodes is the set of nodes to added
15) Figure 3, 4, 5, and 6 have too small font.
16) Line 240: Should this read "Figure 4"?
17) Line 242: figure, When
18) Figure 4 caption: You could mention the training set ratio of 10%.
19) Line 265 and 267: italics d
20) Line 282: "Large nodes"? "After the number of nodes and edges in the network increases, some decision nodes need to command and communicate with more nodes, becoming large nodes in the network." Can you explain and discuss this point in more detail?
21) Figure 6 has the same legends in the left and right figures. Please check these.
22) Please check the statements (and wordings) in lines 291-296.
23) Line 308: [insert article or supplementary material here].
There are some typos in the text (see the list above).
Round 2
Reviewer 1 Report
I accept the answers provided by the authors.
Reviewer 2 Report
The authors have answered my questions.
The English language is fine.